# Domain Adaption for Homogenizing CT Scans using Auto-Encoders for Cross-Dataset Medical Image Analysis

**Mohammadreza Amirian**[*1,2]                                          AMIR@ZHAW.CH
[1] *ZHAW School of Engineering, 8400 Winterthur, Switzerland*
[2] *Ulm University, Institute of Neural Information Processing, 89081 Ulm, Germany*

**Javier A. Montoya-Zegarra**[*1]                                       MONY@ZHAW.CH
**Ahmet Selman Bozkir**[*1]                                             BOZK@ZHAW.CH
**Marco Calandri**[3]                                       MARCO.CALANDRI@UNITO.IT
[3] *University of Turin, Department of Oncology, 10124 Turin, Italy*

**Friedhelm Schwenker**[2]                             FRIEDHELM.SCHWENKER@UNI-ULM.DE
**Thilo Stadelmann**[1,4]                                               STDM@ZHAW.CH
[4] *Fellow, ECLT European Centre for Living Technology, 30123 Venice, Italy*

**Editors:** Under Review for MIDL 2021

## Abstract

Medical imaging research profits from data unification and homogenization methods to merge global datasets in order to reduce annotation effort and improve generalization of trained models to unseen datasets. In this paper, we explicitly address dataset variability using two public datasets and propose an architecture that aims at erasing the differences in CT scans from different sources while simultaneously introducing only minimal changes through leveraging the idea of deep auto-encoders. The proposed trainable prepossessing architecture (PrepNet) ($i$) is jointly trained on the SARS-COVID-2 and UCSD COVID-CT datasets and ($ii$) maintains discriminant features for downstream diagnosis.

**Keywords:** Adaptive preprocessing, domain adaptation, auto-encoder

## 1. Introduction

A major challenge in rolling out machine-learned models to a broad user base is the variability of data encountered in the real world. Models can only be expected to work well on data of similar distribution as has been used for training, but ubiquitously, differences e.g. in the image acquisition setup hinder the applicability of a once developed model in novel settings. This paper uses the example of the negative effects of such failure to adapt between different datasets in the context of COVID-19 diagnosis.

We address domain adaptation of medical image analysis methods by proposing a CNN for preprocessing 2D CT scans: the model is trained to fool a classifier that discriminates between various CT scanning datasets, thus aiming to remove the cross-dataset variability. We evaluate the performance of the suggested method on the exemplary use case of predicting COVID-19 positive cases, due to the global variability in respective datasets and the availability of plenty of opportunities to compare. The methodology is inspired by generative adversarial learning (Schmidhuber, 2020).Our contribution is twofold: ($i$) we propose a novel trainable preprocessing CNN architecture with a dual training objective that is capable of equalizing the variability of different CT-scanner technologies in the image domain (*PrepNet*), see Figure 1 (right); ($ii$) we validate this model by showing the transferability of its diagnostic capabilities between different CT technologies based on common public datasets.

---

[*] Contributed equally

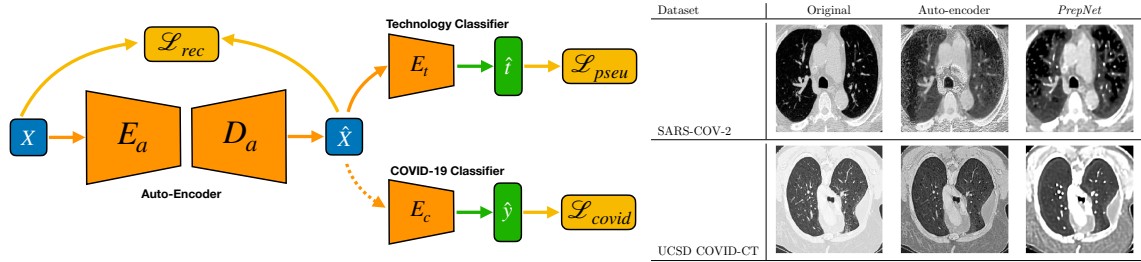

Figure 1: Left: The architecture of our proposed *PrepNet* model consists of three main modules: (*i*) an auto-encoder that acts as a CT cross-dataset homogenizer; (*ii*) a multi CT-technology classifier; and (*iii*) a COVID-19 binary classifier. Right: Images from the used datasets with different prepossessing methods applied.

## 2. Methodology

In this section, we give details of our *PrepNet* model in terms of network architecture, core modules, and loss functions. The architecture of our model is presented in Figure 1 (left). For a group of $\mathcal{N}$ input CT scans $\{\mathcal{X}^n\}_{n=1}^{\mathcal{N}}$, coming from different datasets, our model map the input to a latent space through an auto-encoder and reconstructs the original CT scans $\{\hat{\mathcal{X}}^n\}_{n=1}^{\mathcal{N}}$. The reconstructed CT scans are next fed into a dataset classification branch. The dataset classifier branch uses pseudo-labels for discriminating among different datasets, and the auto-encoder is simultaneously trained in an adversarial way to generate CT scans which fool the dataset classifier and minimizing its accuracy. Once these models are trained end-to-end, the reconstructed CT scans are fed into a COVID-19 classifier which is trained directly on the reconstructed (preprocessed) CT-scans. The COVID-19 classification branch is responsible for the classification of patients into positive and negative cases. The complete network model with its main modules are described as follows:

**Auto-Encoder Module:** We feed a CT scan image $\mathcal{X}^n$ into our auto-encoder ($E_a$ and $D_a$) and obtain a reconstructed version $\hat{\mathcal{X}}^n$ given by $\hat{\mathcal{X}}^n = D_a(E_a(\mathcal{X}^n))$. The encoder $E_a$ is based on the standard classification network VGG16, chosen for its simplicity, whilst the decoder $D_a$ is the mirrored version of the VGG16. We add skip-connections from $E_a$ to $D_a$ and build a U-net architecture to recover the spatial information loss during the down-sampling operations.

**Dataset Classifier Module:** The CT technology classifier $E_t$ receives as input the reconstructed CT scan $\hat{\mathcal{X}}^n$ from the auto-encoder and feeds it into an encoder branch $E_t(\hat{\mathcal{X}}^n)$ that classifies the CT dataset/vendor. $E_t$ relies on the VGG19 architecture.

**COVID-19 Classifier Module:** The COVID-19 classifier $E_c$ is also based on the VGG19 architecture. Given a reconstructed CT scan $\hat{\mathcal{X}}^n$, it outputs COVID vs. non-COVID predictions, i.e. $E_c(\hat{\mathcal{X}}^n)$.

**Loss Functions:** The complete loss function of *PrepNet* is based on the three terms presented in Figure 1 (left). It comprises a reconstruction loss $\mathcal{L}_{rec}$ and two classification losses $\mathcal{L}_{pseu}$ and $\mathcal{L}_{covid}$ to optimize data source prediction and COVID diagnosis:

$$\mathcal{L}_{total} = \mathcal{L}_{rec} + \mathcal{L}_{pseu} + \mathcal{L}_{covid} \tag{1}$$

The final goal of the preprocessing network and training process is removing datasets' variability (minimizing $\mathcal{L}_{pseu}$) with minimum changes to input CT scans (optimizing $\mathcal{L}_{rec}$) while maintaining the necessary information for diagnosis (through $\mathcal{L}_{covid}$).

| Test dataset → Dataset portion | SARS-COV-2 | | | | UCSD COVID-CT | | | | Within Test Average | Cross-Dataset Average |
|---|---|---|---|---|---|---|---|---|---|---|
| | Train | Validation | Test | Average | Train | Validation | Test | Average | | |
| Training dataset ↓ | Original | | | | | | | | | |
| SARS-COV-2 | 0.9533 | **0.9366** | 0.8303 | - | 0.4193 | 0.3310 | 0.4813 | 0.4105 | **0.7890** | 0.4503 |
| UCSD COVID-CT | 0.5084 | 0.4848 | 0.4770 | 0.4901 | **0.8101** | 0.6183 | **0.7478** | - | (Baseline) | (Baseline) |
| | Auto-Encoder | | | | | | | | | |
| SARS-COV-2 | 0.9414 | 0.8980 | 0.6574 | - | **0.5159** | 0.5006 | **0.5041** | **0.5069** | 0.7026 | **0.5209** |
| UCSD COVID-CT | **0.5456** | **0.5173** | **0.5421** | **0.5350** | 0.7934 | **0.7186** | 0.7337 | - | (−8.64%) | (+7.06%) |
| | *PrepNet* | | | | | | | | | |
| SARS-COV-2 | **0.9786** | 0.8157 | **0.8352** | - | 0.5054 | **0.5094** | 0.4835 | 0.4994 | 0.7721 | 0.5002 |
| UCSD COVID-CT | 0.5010 | 0.4173 | 0.5037 | 0.4740 | 0.7279 | 0.6390 | 0.7090 | - | (−1.69%) | (+4.99%) |

Table 1: Cross-dataset validation results using original and preprocessed CT scans.

## 3. Experimental Results

The within- and cross-dataset performances of the proposed preprocessing schemes is evaluated on SARS-COV-2 (Soares and Angelov, 2020) and UCSD COVID-CT (Zhao et al., 2020) public datasets and presented in Table 1. In order to monitor possible overfitting, we report the hold out test set performance when training on the same dataset. The cross-dataset performance is evaluated by measuring the balanced accuracy of the models trained on one dataset and tested on all three (train, validation and test) sets of the other dataset. We report results using the balanced accuracy of COVID-19 diagnosis to minimize the effect of class imbalance and label distribution in two datasets. For each dataset, we report on the columns the results on different dataset splits: train, validation, and test. In the rows, we present the datasets used for training. Furthermore, we group the results by preprocessing techniques. The first group of results are related to the VGG19 classifier trained and evaluated on the original CT scans undergo fixed conventional preprocessing steps (histogram equalization and normalization). The second group of results is related to the auto-encoder alone trained on training sets of both datasets in a self-supervised manner to minimize the reconstruction loss. The third group of results relate to full *PrepNet* preprocessing before training the classifiers depicted in Figure 1 (right). The results in Table 1 demonstrate that the average cross-dataset performance (over all dataset splits) of models trained on original data increases by 7.06pp after using the pure auto-encoder model, and by 4.99pp through applying *PrepNet*. However, the average test accuracy for within-dataset evaluation declines by 8.64pp and 1.69pp after applying the baseline auto-encoder or PrepNet, respectively.

## 4. Conclusions and Future Works

This paper presented an architecture to unify several CT scan datasets concerning varying image acquisition circumstances such as CT scanner vendors through a trainable and adaptive preprocessing network that removes such specificities from the images themselves. We presented preliminary experimental results demonstrating the applicability of the method on two publicly available datasets. The proposed *PrepNet* improves the cross-dataset balanced accuracy at the expanse of a decline in the within-dataset test performance. Further optimizing *PrepNet* to boost both within- and cross-dataset performances, using more datasets, and applying the methodology to other diseases are amongst the future works.

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
