# OpenReview forum: "Domain Adaption for Homogenizing CT Scans using Auto-Encoders for Cross-Dataset Medical Image Analysis"
_MIDL.io/2021/Conference/Short — Submitted to MIDL 2021_

### Official Review · Reviewer_JTSd · 2021-05-04

**Confidence:** 5
**Final Rating:** 2

**Summary:**

The paper presents a deep learning architecture (PrepNet) that acts as a cross dataset homogenizer. The architecture is composed of three different parts an autoencoder module that reconstructs the given input trained in an adversarial way, a dataset classifier module that identifies the dataset origin of the inputs, and a classification module that identifies the presence or not of COVID-19. Experiments in two different 2D COVID-19 datasets show the better performance of the presented architecture compared to other two other baselines.


**Strengths:**

What are the strengths of the paper? Clearly explain why these aspects of the paper are valuable. Rate the paper on the basis of its scientific merits and potential value to the community. I
- The topic of the paper is interesting and appropriate for MIDL 2021
- The paper is easy to follow.


**Weaknesses:**

- The paper in this form misses some important information about the technical details that are in my opinion important to ensure reproducibility and proper understanding of the different components. For example, the loss functions used in eq. 1 are not defined properly and it is really unclear on which components they are applied (e.g. the reconstruction loss).
- The provided results are a bit mixed and it is not very clear which component of the proposed method contributes the most to the performance of the architecture.
- I am really skeptical about the use of skip connections on a reconstruction autoencoder. By using the skip connections the authors pass almost most of the information needed for the reconstruction directly on the last layer of the autoencoder.


**Deanonymize Review:**

no

**Detailed Comments:**

- I think it is important to define the type of losses used on the proposed formulation. For example, for reconstruction did the authors use the L1 or L2 norm? For the classification, did the authors use the binary cross entropy, focal loss, etc? It is not easy to review the technical components of the proposed pipeline in this version of the manuscript.
- All the implementation details, time, parameter complexity are missing.
- In Figure 1 I think the authors should change the window intensities to be stable between the different visualized slides.
- How the reconstructed images are validated? Are there some failing cases?
- Maybe additional metrics such as the F1 score could be interesting to be presented.


**Justification Of The Rating:**

Overall, I think the paper is interesting, however, I think that important technical details are missing making its reproducibility difficult and proper understanding of the paper, while I have some questions about different components of the approach such as the use of skip connections and the performance influence of the different branches.


**Paper Type:**

methodological development

**Special Issue:**

no

---

### Official Review · Reviewer_Gszq · 2021-05-07

**Confidence:** 4
**Final Rating:** 2

**Summary:**

This paper suggests a GAN-like architecture for homogenizing data cross sites and scanners, specifically aiming to erase "the differences in CT scans from different sources while simultaneously introducing only minimal changes through leveraging the idea of deep auto-encoders". The authors use two datasets (SARS-COVID-2 and UCSD COVID-CT) for this work, which involves homogenizing data and producing a DNN-based diagnosis.

**Strengths:**

- The problem of CT data homogenization is interesting and well motivated
- The use of a GAN-like architecture to achieve this is particularly interesting
- The architecture is well described and visualized in Fig. 1.

**Weaknesses:**

- The training scheme is not clear. How low does L_rec have to be? If L_rec = 0, will the technology classifier always succeed? This scheme seems distinct from GANs used in other applications (e.g. RGB image synthesis), where improving the reconstruction/synthesis actually makes it harder for the discriminator to achieve high accuracy.
- The results in Table 1 are not clear. Shouldn't we compare training with one or both datasets? This is also not well explained in the Experimental Results section. What is the ultimate measure of homogenization?
- The visual similarity is not evaluated or reported, e.g. using SSIM or MAE. This seems to be an important point, since otherwise a single DNN could be trained using both datasets.

**Deanonymize Review:**

no

**Detailed Comments:**

- The practical applicability of a COVID classifier from CT is not clear.
- Details of the dataset and the input/outputs of the COVID classifier are not clear. Is it just positive/negative?
- Why do we want the images to be homogenized anyway? Why not just homogenize the classifier w.r.t. different data types if thats what is being evaluated at the end?
- It is unclear whether the homogenization of CT scans was actually achieved. (Per the introduction, this is independent of the ultimate COVID classifier in the last stage.)

**Justification Of The Rating:**

The presented technique is potentially interesting, but the paper does not provide numerical evidence that homogenization is achieved. The results of the COVID classifier indicate that classification performance is improved/maintained, but it is not clear why this is the ultimate measure of homogenization. In addition, the training scheme is not adequately explained or justified, since it deviates from traditional GAN architectures.

**Paper Type:**

both

**Special Issue:**

no

---

### Meta-Review · Area_Chair_FLCP · 2021-05-09

**Recommendation:** Reject
**Confidence:** 5

**Metareview:**

The reviewers remark missing details concerning motivation, reproducibility, and interpretation of the findings.

---

### Decision · Program_Chairs · 2021-05-11

Reject